# Differences in Cerebral Glucose Metabolism in ALS Patients with and without *C9orf72* and *SOD1* Mutations

**DOI:** 10.3390/cells12060933

**Published:** 2023-03-18

**Authors:** Joke De Vocht, Donatienne Van Weehaeghe, Fouke Ombelet, Pegah Masrori, Nikita Lamaire, Martijn Devrome, Hilde Van Esch, Mathieu Moisse, Michel Koole, Patrick Dupont, Koen Van Laere, Philip Van Damme

**Affiliations:** 1Division of Psychiatry, Division of Neurology, University Hospitals Leuven, VIB-KULeuven Center for Brain & Disease Research, Laboratory of Neurobiology, Department of Neurosciences, Leuven Brain Institute (LBI), Katholieke Universiteit Leuven, 3000 Leuven, Belgium; 2Department of Radiology and Nuclear Medicine, Ghent University Hospital, 9000 Ghent, Belgium; 3Division of Neurology, University Hospitals Leuven, VIB-KULeuven Center for Brain & Disease Research, Laboratory of Neurobiology, Department of Neurosciences, Leuven Brain Institute (LBI), Katholieke Universiteit Leuven, 3000 Leuven, Belgium; 4Department of Imaging and Pathology, Nuclear Medicine and Molecular Imaging, Katholieke Universiteit Leuven, 3000 Leuven, Belgium; 5Center for Human Genetics, University Hospitals Leuven, 3000 Leuven, Belgium; 6VIB-KU Leuven Center for Brain & Disease Research, Laboratory of Neurobiology, Department of Neurosciences, Leuven Brain Institute (LBI), Katholieke Universiteit Leuven, 3000 Leuven, Belgium; 7Laboratory of Cognitive Neurology, Department of Neurosciences, Leuven Brain Institute (LBI), Katholieke Universiteit Leuven, 3000 Leuven, Belgium

**Keywords:** genetic ALS, FDG PET, *C9orf72*, *SOD1*

## Abstract

Amyotrophic lateral sclerosis (ALS) is characterized by progressive loss of upper and lower motor neurons. In 10% of patients, the disorder runs in the family. Our aim was to study the impact of ALS-causing gene mutations on cerebral glucose metabolism. Between October 2010 and October 2022, 538 patients underwent genetic testing for mutations with strong evidence of causality for ALS and ^18^F-2-fluoro-2-deoxy-D-glucose-PET (FDG PET), at University Hospitals Leuven. We identified 48 *C9orf72*-ALS and 22 *SOD1*-ALS patients. After propensity score matching, two cohorts of 48 and 21 matched sporadic ALS patients, as well as 20 healthy controls were included. FDG PET images were assessed using a voxel-based and volume-of-interest approach. We observed widespread frontotemporal involvement in all ALS groups, in comparison to healthy controls. The degree of relative glucose metabolism in *SOD1*-ALS in motor and extra-motor regions did not differ significantly from matched sporadic ALS patients. In *C9orf72*-ALS, we found more pronounced hypometabolism in the peri-rolandic region and thalamus, and hypermetabolism in the medulla extending to the pons, in comparison to matched sporadic ALS patients. Our study revealed *C9orf72*-dependent differences in glucose metabolism in the peri-rolandic region, thalamus, and brainstem (i.e., medulla, extending to the pons) in relation to matched sporadic ALS patients.

## 1. Introduction

The multifaceted clinical syndrome of amyotrophic lateral sclerosis (ALS) presents with rapidly progressing neurodegeneration in motor and extra-motor regions and is usually fatal within 2–5 years after disease onset [1]. The disorder is characterized by varying phenotypic manifestations with degeneration of both upper motor neurons (UMN) in the motor cortex and lower motor neurons (LMN) in the brainstem and spinal cord [2]. This leads to progressive muscle weakness and paralysis. ALS has an average age of onset of 55–65 years, and an incidence of 1.5–2.7 per 100,000 inhabitants per year [1].

Even though most patients with ALS have a form of the condition that is described as sporadic, which means there is no apparent family history of the disorder, approximately 10% of patients have a familial form of ALS (fALS). In about 60–80% of fALS and 10% of sporadic ALS, gene mutations of large effect, with autosomal dominant inheritance, are present [1,2].

The most prevalent identified cause of ALS found in patients of European descent is a hexanucleotide repeat expansion (HRE) in the *C9orf72* gene (OMIM 614260), accounting for 40–60% of fALS cases and 5–10% of apparently sporadic ALS cases [3,4]. *C9orf72*-associated ALS (*C9orf72*-ALS) is characterized by a wide range of phenotypic presentations, including a pure frontotemporal dementia (FTD) phenotype. Compared to sporadic ALS, ALS patients with a *C9orf72* HRE more frequently present with bulbar onset and with cognitive and behavioral deficits [5]. Site of onset, age at onset, clinical presentation, disease progression, and (the extent of) cognitive impairment varies considerably within and across families [6,7,8].

Variants in the Cu/Zn superoxide dismutase-1 gene (OMIM 147450) are the second most common cause of ALS and account for 12–24% of fALS cases and 1% of apparently sporadic ALS cases, in populations of European descent [9,10,11,12,13,14,15]. Thus far more than 200 pathogenic mutations are reported in the Human Genome Mutation Database [10,16,17]. From a clinical point of view, *SOD1*-associated ALS (*SOD1*-ALS) is characterized by marked phenotypic variability. Some monogenetic causes are associated with a shorter survival (e.g., Ala5Val mutation), while other variants more commonly found in Western and Southern Europe are associated with a slower disease progression [10,11,12,13]. Cognitive involvement in *SOD1*-ALS is seldomly reported. A recent study however suggested that behavioral changes may be more common than previously thought among patients harboring *SOD1* variants [18].

Other “major” ALS-causing gene mutations are mutations in TAR DNA-binding protein 43 (*TARDBP*-ALS) and in fused in sarcoma (*FUS*-ALS), each accounting for 1–5% of fALS cases [19,20].

Brain imaging is increasingly used to measure ALS-associated changes in vivo. Previous studies provided evidence of the diagnostic value of ^18^F-2-fluoro-2-deoxy-D-glucose-PET (FDG PET) in ALS, as peri-rolandic and variable prefrontal hypometabolism is found in most patients [21,22,23,24]. In addition, it appears to correlate with (the extent of) cognitive involvement and the presence of concomitant FTD, and has associated prognostic value [25].

Little research, however, has been done into the cerebral glucose metabolic correlates of monogenic forms of ALS. FDG PET studies that investigated the metabolic correlates associated with a *C9orf72* HRE reported inconsistent findings concerning the extent of cortical involvement in *C9orf72*-ALS [21,26,27,28]. One study involving *C9orf72*-ALS patients described extensive and more widespread cortical hypometabolism in *C9orf72*-ALS than in sporadic ALS [26]. However, this finding was not replicated in two other studies [21,28]. To our knowledge, only one study reported on glucose metabolic changes in *SOD1*-ALS [29]. This FDG PET study by Canosa et al. described a cluster of relative hypermetabolism in the peri-rolandic region in *SOD1*-ALS patients in relation to sporadic ALS patients and healthy controls [29].

In this study, we investigated the glucose metabolic pattern associated with the two most prevalent monogenic causes of ALS in a large cohort, with a particular focus on comparing patterns of significant relative hypo- and hypermetabolism between familial and sporadic ALS.

## 2. Materials and Methods

The local Ethical Committee of the University Hospitals Leuven approved this study (s50354). Informed consent was obtained from all subjects in accordance with the Declaration of Helsinki.

### 2.1. Subjects

#### 2.1.1. Healthy Controls

The healthy control population (HC) (*n* = 20; mean age ± SD, 62.4 ± 6.4 y, 8M/12F) was the same as that used in a previous FDG PET study [21].

#### 2.1.2. Patients

A total number 538 incident patients were recruited from the Neuromuscular Clinic in the University Hospitals Leuven between October 2010 and October 2022. All patients were diagnosed by an experienced neurologist specialized in neuromuscular disorders, using the revised El Escorial and Awaji-Shima criteria [30], and underwent a diagnostic ^18^F-2-fluoro-2-deoxy-D-glucose-PET (FDG PET).

Out of the 538 patients, we considered 533 participants for this study. We excluded five participants because of comorbidities (brain cyst (*n* = 1), spastic paraparesis (*n* = 2), cerebrovascular accident (*n* = 2)).

In 84 patients with ALS, a pathogenic mutation was found, known to be associated with ALS. Most commonly, we found a *C9orf72* HRE (*n* = 48, 57%), followed by a mutation in the *SOD1* gene (*n* = 22; 26%), a mutation in the *TARDBP* gene (*n* = 7; 8%), a mutation in the *FUS* gene (*n* = 5; 6%), and lastly, a dual *C9orf72* and *TARDBP* gene mutation (*n* = 2; 2%). Given the limited sample size of the three latter groups, they were not considered for further analyses.

### 2.2. Propensity Score Matching

The aim of this approach was to balance the selected covariance between the cohorts of observational data to minimize the effect of confounding factors. From an initial group of 449 sporadic ALS patients, we performed propensity score matching (PSM) [31,32] using a logistic regression model based on the following six variables: age at FDG PET, sex, King’s disease stage, diagnostic delay, the extent of lower vs. upper motor neuron involvement at diagnosis, and scanner type. Case-control matching for scanner type was performed, such that scanner type would not bias observed differences in the cerebral glucose metabolism. One-to-one pair matching was performed without replacement, using a caliper of 0.20.

The algorithm matched 48 *C9orf72*-associated ALS patients and 21 out of 22 *SOD1*-associated ALS patients to an equal number of sporadic ALS patients. One patient displayed a distinct phenotype of rapidly progressive respiratory onset ALS at a young age. A similar phenotype has not been observed before in our cohort of sporadic ALS patients.

The total study population consisted of 48 patients with *C9orf72*-associated ALS (*C9orf72*-ALS), 48 sporadic ALS patients matched to the *C9orf72*-ALS patient cohort (*^C9orf72^*^-matched^sALS), 22 patients with *SOD1*-associated ALS (*SOD1*-ALS), 21 sporadic ALS patients matched to the *SOD1*-ALS cohort (*^SOD1^*^-matched^sALS), and 20 healthy controls.

### 2.3. Assessment of Clinical Functioning

The clinical assessment at diagnosis and revised Amyotrophic Lateral Sclerosis Functional Rating Scale (ALSFRS-r) were utilized to perform King’s clinical staging [33].

### 2.4. Image Analysis

#### 2.4.1. PET Acquisition and Reconstruction

All scans were acquired during the diagnostic work-up of patients. All subjects fasted at least 6 h before [^18^F] FDG injection. FDG PET scans were acquired using an ECAT HR+ camera (Siemens), a Biograph 16 HiRez PET/CT camera (Siemens), or a Biograph 40 Truepoint PET/CT camera (Siemens). In all cases, a bolus of [^18^F] FDG was injected intravenously under standard conditions: lying in a dimly lit, quiet room with eyes and ears open.

PET acquisitions were performed in list mode thirty minutes after injection. Images were acquired dynamically for 30 min on the ECAT HR+ (3D mode) and for 15–20 min on the HiRez and Truepoint (3D and listmode). During the acquisition, the patient’s head was immobilized with a vacuum pillow or headholder. All images were corrected for attenuation, scatter, and decay.

On the ECAT HR+, images were corrected for attenuation with the attenuation map, acquired with a transmission scan using ^68^Ge/^68^Ga rod sources, and projection data were reconstructed using 3-dimensional filtered back projection with a Hanning post-filter. On the HiRez and Truepoint, images were reconstructed using ordered-subset expectation maximization (OSEM) with 5 iterations over 8 subsets and 3 iterations over 21 subsets, respectively, with an 8 mm Gaussian kernel.

#### 2.4.2. FDG PET Image Analysis

All scans were spatially normalized to Montreal Neurological Institute (MNI) space using a FDG PET template (SPM12; Welcome Trust Centre for Neuroimaging), implemented in Matlab (R2020b, The MathWorks Inc., Natick, MA, USA), and nonrigid registration with 16 iterations, in voxels of 2 mm × 2 mm × 2 mm. Isotropic Gaussian smoothing with full width at half maximum of 8 mm was performed, followed by proportional scaling to the mean uptake value within a brain mask, obtained by applying a relative threshold masking of 0.8 in the healthy control group.

Voxel-based findings were corroborated with an atlas-based volume-of-interest (VOI) analysis using the N30R83 Hammers probabilistic atlas and the Automated Anatomical Labeling (AAL) single-subject atlas implemented in PNEURO (PMOD, v3.7, PMOD Inc., Zurich, Switzerland) [34]. The AAL single-subject atlas was used to delineate the midbrain, pons, and medulla, enabling a more detailed delineation of the entire brainstem [35]. Standardized uptake value ratio (SUVR) values were calculated, by scaling the mean uptake in VOIs to the average grey matter activity to obtain relative metabolic activity.

### 2.5. Statistical Analysis

General statistics were done using SPSS 28.0 (SPSS Inc. Chicago, IL, USA). Data are presented as median (interquartile range). Normality of the distributions of the clinical data was assessed with a Shapiro–Wilk test (α = 0.05). For statistical comparisons, we used the χ2 test for categorical variables, or Kruskal–Wallis test for continuous variables. Survival was estimated using the Kaplan–Meier method and compared using log-rank tests, with Bonferroni-corrected post hoc pairwise comparisons.

For the voxel-based analyses, we used an analysis of variance (ANOVA) model, using the SPM12 toolbox, implemented in Matlab, to assess the effect of genetic status on cerebral glucose metabolic uptake. Statistical significance for the voxel-based analyses was set at p_uncorr_ < 0.001, combined with a p_FWE_ < 0.05 at the cluster level, unless otherwise specified.

VOI-based independent Mann–Whitney U tests were conducted to determine if there were statistically significant differences in relative metabolic activity between *C9orf72*-ALS and *^C9orf72^*^-matched^sALS, and between *SOD1*-ALS and *^SOD1^*^-matched^sALS. We also compared each disease group to the healthy control group. We evaluated 37 regions, and the Benjamini–Hochberg procedure was used to control the false discovery rate (FDR) at 0.05 [36].

## 3. Results

### 3.1. Demographics and Clinical Characteristics

In total, 48 *C9orf72*-associated ALS patients (*C9orf72*-ALS), 22 *SOD1*-associated ALS patients (*SOD1*-ALS), and 69 matched sporadic ALS patients (*^C9orf72^*^-matched^sALS, *n* = 48; *^SOD1^*^-matched^sALS, *n* = 21) were included, with a median age of 59 years (range, 29–84 years). The median injected [^18^F] FDG activity was 149 MBq (range, 106–283 MBq).

As we know that the disease progression rate between *C9orf72*-ALS and *SOD1*-ALS differs significantly, we employed propensity score matching to obtain “artificial sporadic ALS control cohorts”, matched to our *C9orf72*-ALS and *SOD1*-ALS cohorts. A comparative analysis did reveal a statistically significant difference in diagnostic delay (*p* < 0.001) between different ALS disease groups. Detailed demographics and clinical characteristics of each group are reported in Table 1.

As the disease progression rate between *C9orf72*-ALS and *SOD1*-ALS differs substantially, we compared survival rate between all four groups. A log rank test (Mantel–Cox) revealed that the survival distribution for the four ALS patient groups was significantly different, χ2(3) = 11.130, *p* = 0.011 (Figure 1). Pairwise comparisons revealed a significant difference between the unmatched genetic groups, *SOD1*-ALS and *C9orf72*-ALS, χ2(1) = 13.922, *p* = 0.0002 (Figure 1, Table 1). *SOD1*-ALS patients had a median survival time of 114 months, while *C9orf72*-ALS patients had a median survival time of 29 months and matched sporadic ALS patients had a median survival time of 22 months (*^SOD1^*^-matched^sALS) and 32 months (*^C9orf72^*^-matched^sALS).

A log rank test revealed a significantly different distribution in survival for the different ALS groups, χ2(3) = 11.130, *p* = 0.011. Log rank pairwise comparisons were done with a Bonferroni correction, with statistical significance accepted at the *p* < 0.0125 level. There was a significant difference in survival distribution for *SOD1*-ALS and *C9orf72*-ALS patients (*p* = 0.0002).

### 3.2. Cerebral Glucose Metabolic Changes between Different Genotypes

Voxel-based comparative analyses between *C9orf72*-ALS and *SOD1*-ALS patients revealed clusters of relative hypermetabolism in the brainstem (i.e., pons, extending to the midbrain) and cerebellum of *C9orf72*-ALS patients and relative hypometabolism bilaterally in the basal ganglia of *SOD1*-ALS patients (Figure 2 and Appendix A).

The results of the VOI-based Mann–Whitney U test (Table 2, Appendix A) corroborated our findings at the voxel level in the basal ganglia (i.e., lentiform nucleus), brainstem (i.e., pons and midbrain), and cerebellum. In addition, the VOI-based comparative analysis revealed a significant difference in FDG uptake between *C9orf72*-ALS and *SOD1*-ALS patients in the peri-rolandic region, posterior cingulate gyrus, thalamus, and medulla.

### 3.3. Differences in Cerebral Glucose Metabolism between ALS Groups and Healthy Controls

We compared all ALS groups to a group of healthy controls. This revealed significantly reduced glucose metabolism in fronto-insular, temporal, parietal, and occipital cortices, the amygdala, basal ganglia, and thalamus in all ALS groups (Appendix A, Appendix A). Hypermetabolism was observed in the cerebellum in *C9orf72*-ALS, *SOD1*-ALS, and *^C9orf72^*^-matched^sALS patients and in the brainstem in *C9orf72*-ALS patients.

### 3.4. Effect of Genetic Status on Cerebral ^18^F FDG Uptake

Voxel-based comparative analyses between *C9orf72*-ALS and *^C9orf72^*^-matched^sALS patients, and between *SOD1*-ALS and *^SOD1^*^-matched^sALS patients, revealed two significant clusters of relative hypometabolism and one cluster of relative hypermetabolism in *C9orf72*-ALS patients in relation to *^C9orf72^*^-matched^sALS patients.

This study demonstrated that *C9orf72*-ALS was associated with reduced glucose metabolism in the peri-rolandic region, extending to the precuneus, and the right thalamus, as well as increased glucose metabolism in the brainstem (i.e., medulla, extending to the pons), in relation to the *^C9orf72^*^-matched^sALS group (Figure 3; Appendix A).

Voxel-based analyses between *SOD1*-ALS and *^SOD1^*^-matched^sALS groups revealed no significant clusters of relative hypo- or hypermetabolism. The results of the VOI-based Mann–Whitney U test (Table 3 and Table 4, Appendix A) corroborated our findings at the voxel level.

## 4. Discussion

The primary objective of this study was to investigate the effect of major gene mutations causative for ALS on cerebral glucose metabolism. The considerable variability in the phenotypic expression of ALS that is reflected in familial and sporadic ALS [8] makes comparative analyses between subgroups of ALS patients challenging.

To ensure that any differences in cerebral glucose metabolism between genotypes and sporadic ALS patients were not driven by population diversity, we matched *C9orf72*-ALS and *SOD1*-ALS patients to our cohort of sporadic ALS patients, using propensity score matching.

Comparing two genetic subtypes of ALS (i.e., *C9orf72*-ALS and *SOD1*-ALS) and their respective matched cohorts of sporadic ALS patients (i.e., *^C9orf72^*^-matched^sALS and *^SOD1^*^-matched^sALS) readily revealed more pronounced hypometabolism in the peri-rolandic region, extending to the precuneus, and thalamus, as well as hypermetabolism in the brainstem (i.e., medulla, extending to the pons), in *C9orf72*-ALS, when compared to *^C9orf72^*^-matched^sALS.

### 4.1. The Effect of C9orf72 Status on Cerebral Glucose Metabolism

Previous research has produced contradictory research findings about the glucose metabolic signature in patients with a *C9orf72* HRE [21,26,27,28]. The study by Van Laere et al. produced no significant differences between *C9orf72*-ALS and sporadic ALS [21], while the study by Diehl-Schmid et al. reported reduced glucose metabolism confined to the thalamus [28] in *C9orf72*-FTD and sporadic FTD.

Another study by Cistaro et al., on the other hand, described widespread reduced glucose metabolism in the left frontal and superior temporal cortices, the bilateral anterior and posterior cingulate cortex, insula, caudate, and thalamus, in *C9orf72*-ALS in comparison to sporadic ALS, and relative hypermetabolism in the midbrain, bilateral occipital cortex, globus pallidus, and left middle and inferior temporal cortices [26].

To overcome the challenge of selection bias that might occur due to the nature of retrospective studies and to minimize the effect of confounding factors, we studied the cerebral glucose metabolic pattern of 48 ALS patients with a *C9orf72* HRE in relation to a group of matched sporadic ALS patients, using PSM.

The clusters of significantly reduced glucose metabolism in the peri-rolandic region, extending to the precuneus, and thalamus suggest that both the thalamus and the peri-rolandic region play a key role in *C9orf72*-mediated disease [26,27,28,37]. Reduced FDG uptake in *C9orf72*-ALS patients can reflect multiple neurobiological underpinnings, such as reduced excitability [38], synapse loss [39], and neurodegeneration. Mitochondrial dysfunction [40] may also contribute to the observed changes, as a recent study employing the [^18^F]2-tert- butyl-4-chloro-5-2H- pyridazin-3-one (BCPP-EF) radioligand for mitochondrial complex demonstrated that mitochondrial complex I reduction can be linked to a range of downstream neurodegenerative processes, such as hypometabolism. Follow-up PET studies with [^18^F]BCPP-EF and [^18^F]FDG would enable in vivo investigation of dysfunction of mitochondria and its relation to hypometabolism in *C9orf72*-ALS [41].

Thalamic involvement may not only be one of the earliest features of *C9orf72*-mediated disease, as demonstrated by PET studies in presymptomatic *C9orf72* HRE carriers [37,42], but also one of the regions that is most severely affected by the disorder.

In comparison with healthy controls, we found relative hypermetabolism in the cerebellum, the medial temporal lobe, and the medulla, extending to the pons in *C9orf72*-ALS. Cerebellar and medial temporal hypermetabolism appears relatively prominent but not specific to *C9orf72*-ALS, as we found significant changes in the medulla (extending to the pons) when compared to matched sporadic ALS patients.

The neurobiological processes that underlie hypermetabolism in these regions remain to be elucidated. Various theories have been proposed to explain this observation, such as the normalization procedures used [21], recruitment of neural networks as a localized inflammatory response [21,24,25,43], or even a compensatory neuronal mechanism as the frontal regions (including the primary motor cortex) degenerate [44].

Indeed, unlike previous observations in *C9orf72*-associated FTD [27,28], we observed that the medulla was significantly hypermetabolic in *C9orf72*-ALS patients when compared to *^C9orf72^*^-matched^sALS patients. A human autopsy study already demonstrated that *C9orf72*-ALS cases had more severe microglial pathology in the medulla compared to ALS cases without a *C9orf72* hexanucleotide repeat expansion [45]. Relative hypermetabolism in this region may thus represent neuroinflammation associated with activated astrocytes or microglia [46]

This significant hypermetabolism in the medulla may be associated with a more aggressive disease phenotype in patients with a *C9orf72* hexanucleotide repeat expansion [47,48,49], as a recent study with FDG PET/MRI demonstrated that medulla hypermetabolism is associated with shortened survival [50].

### 4.2. The Effect of SOD1 Status on Cerebral Glucose Metabolism

In our study, we found no significant differences in cerebral glucose metabolism between *SOD1*-ALS patients and matched sporadic ALS patients. Moreover, a comparison to healthy controls revealed a pattern of widespread relative hypometabolism in frontal, temporal, parietal, and subcortical regions (including basal ganglia and thalamus) in *SOD1*-ALS, resembling the glucose metabolic pattern observed in sporadic ALS [21,23,51,52].

Our findings are in line with several postmortem studies that described lesions of the (extrapyramidal) motor and nonmotor systems in *SOD1*-positive cases, including subcortical structures as well as frontotemporal cortices [53,54]. A recent study also revealed that behavioral changes are more commonly found in patients carrying *SOD1* variants than previously thought [18].

In past years, a few in vivo neuroimaging studies have been conducted in *SOD1*-associated ALS. Two volumetric imaging studies demonstrated similar cerebral volumetric measurements between *SOD1*-ALS patients and sporadic ALS patients [55,56]. Two PET imaging studies using [^11^C]Flumazenil and [^11^C]WAY100635 described a more focal loss of GABAergic cortical inhibition and serotonergic binding differences in patients with a *SOD1*^D91A^ mutation than in sporadic ALS patients [57,58]. More recently, a study employing ^11^C-PK11195 PET revealed widespread, diffuse cortical and subcortical increased glial activation in *SOD1*-associated ALS [59], comparably to what was observed in sporadic ALS [60].

A recent FDG PET study by Canosa et al. described a cluster of relative hypermetabolism in the peri-rolandic region in *SOD1*-ALS patients in relation to sporadic ALS patients and healthy controls [29]. Differences found between both FDG PET studies (i.e., our study and the study by Canosa et al. [29]) may be explained by the differences between the *SOD1*-ALS cohorts. A quarter of *SOD1*-ALS patients in the study by Canosa et al. had pure lower motor neuron involvement [29], a phenotype we did not observe in our cohort (Appendix A). Moreover, both cohorts differed in the distribution of genetic variants, as in our cohort, the majority of *SOD1* patients harbored an *SOD1*^D91A^ (45%; *n* = 10) or *SOD1*^G94C^ genetic variant (27%; *n* = 6) (Appendix A).

Differences may also be attributed to the use of propensity score matching to minimize the impact of confounding variables. Unlike sequential bivariate matching, this technique accounts for potential multivariate interactions between the matching variables, which may introduce bias in the participant selection [31].

Nevertheless, large multicenter approaches investigating the cerebral glucose metabolism associated with different pathogenic *SOD1* variants are needed to better understand the variability in disease presentations of different *SOD1* mutations.

Our study has several limitations.

Firstly, subjects were scanned on three different PET cameras. Slight differences associated with CT-based versus 68Ge/Ga-based attenuation correction (especially in infratentorial areas), as well as spatial resolution may have contributed to an increased variance. However, the spatial resolution of the three scanners is highly similar. Nevertheless, case-control matching for scanner type was performed, such that scanner type would not bias observed differences in the cerebral glucose metabolism.

Secondly, the absence of individual MR images for each subject did not allow for partial volume correction. Cerebral glucose metabolism measured by PET may capture both implicit cellular metabolic functions and atrophy. However, our findings on metabolic signatures are in line with those reported in previous studies [21,23,24,26,27,28].

Thirdly, we tried to correct for confounding factors using PSM, but bias exposures are inevitable due to the respective nature of the study. PSM can account for multivariate interactions, minimizes selection error, and is better suited for smaller sample sizes [31,61].

Fourthly, due to the small sample size, we excluded the following carriers of a monogenic mutation causative for ALS in our cohort (*TARDBP*, *n* = 7; *FUS*, *n* = 5; *TARDBP* and *C9orf72*, *n* = 2) from all analyses.

Fifthly, we observed no significant differences between *SOD1*-ALS and sporadic ALS patients. We cannot exclude that between-group differences may have been washed out, due to within-group heterogeneity. This may have also reduced the power of our analyses. However, a post hoc power analysis on our cortical VOIs revealed that there was sufficient power, which suggests that the lack of significant differences may be due to a small effect size rather than to a shortage of power.

Finally, there were some discrepancies between VOI-based Mann–Whitney U tests and our voxel-based findings, possibly due to VOI dilution or our approach to multiple comparisons correction (i.e., familywise error rate versus false discovery rate (FDR)). Due to the lack of contemporaneous MRIs, the VOIS included in this analysis comprise different distributions of tissue classes at the individual level, which may potentially confound the interpretation. Secondly, the FDR is a less conservative approach to multiple comparisons correction and may result in a false-positive finding.

## 5. Conclusions

In conclusion, our findings demonstrate that the distribution of metabolic activity, using FDG PET, in *SOD1*-associated ALS resembles the changes observed in matched sporadic ALS patients, when compared to healthy controls. To enable more sensitive analyses, large multicenter approaches may aid in investigating the cerebral glucose metabolism associated with different pathogenic *SOD1* variants.

Patients with a *C9orf72* HRE, on the other hand, did present with significant differences in FDG uptake in the peri-rolandic region, thalamus, and brainstem (i.e., medulla, extending to the pons) compared to matched sporadic ALS patients and healthy controls. The results from the current study extend previous findings by identifying that, in addition to the thalamus, the peri-rolandic region and medulla play a key role in *C9orf72*-mediated ALS. Our findings support the hypothesis that ALS patients with a *C9orf72* HRE demonstrate more central nervous system involvement than sporadic ALS and *SOD1*-associated ALS patients do.

## Figures and Tables

**Figure 1 cells-12-00933-f001:**
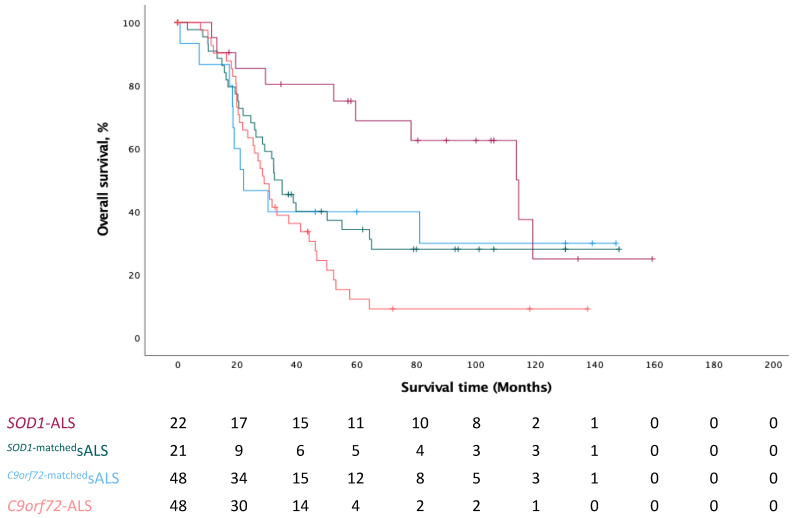
Kaplan–Meier plot of overall survival by ALS group.

**Figure 2 cells-12-00933-f002:**
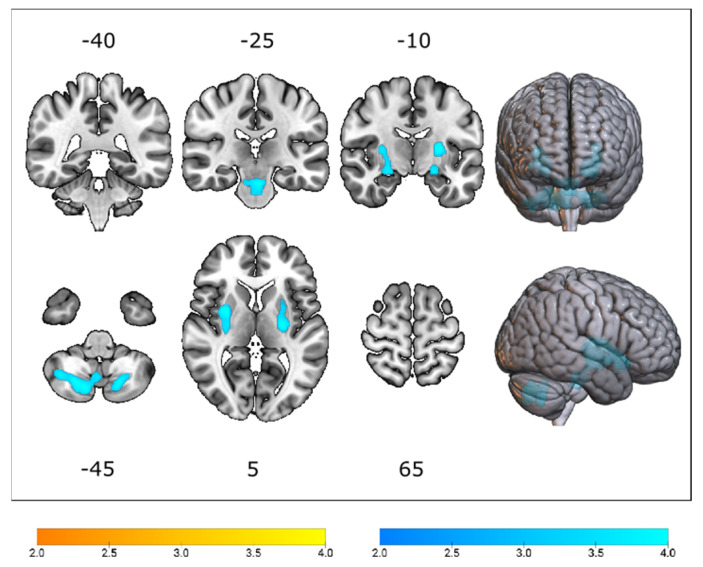
T-statistical map detailing patterns of relative hypometabolism (yellow) and hypermetabolism (blue) in *C9orf72*-ALS patients, when compared to *SOD1*-ALS patients, thresholded at p_uncorr_ < 0.001 at voxel level, and p_FWE_ < 0.05 at cluster level, while correcting for age, sex, scanner type and King’s disease stage. Clusters are overlaid on a T1-weighted template.

**Figure 3 cells-12-00933-f003:**
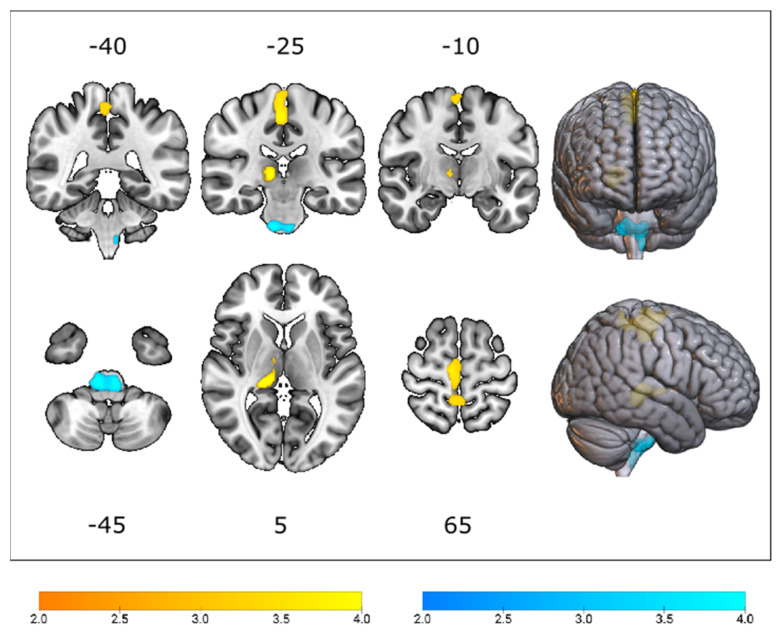
T-statistical map detailing patterns of relative hypometabolism (yellow) and hypermetabolism (blue) in *C9orf72*-ALS, when compared to a group of matched sporadic ALS patients, thresholded at p_uncorr_ < 0.001 at voxel level, and p_FWE_ < 0.05 at cluster level. Clusters are overlaid on a T1-weighted template.

**Table 1 cells-12-00933-t001:** Demographics and clinical characteristics of ALS patient groups (healthy controls, *n* = 20; *C9orf72*-ALS, *n* = 48; *^C9orf72^*^-matched^sALS, *n* = 48; *SOD1*-ALS, *n* = 22; *^SOD1^*^-matched^sALS, *n* = 21). Abbreviations: C9orf72-ALS, C9orf72-associated ALS; CI, confidence interval; IQR, interquartile range; M, months; Md, median; n, number; SOD1-ALS, SOD1-associated ALS; Yr, years.

Md (IQR)	*C9orf72*-ALS	*^C9orf72^*^-matched^sALS	*SOD1*-ALS	*^SOD1^*^-matched^sALS	Test Statistic	*p*-Value
Age at FDG PET (Yr)	58.5 (9.00)	63 (16.75)	54.5 (20.00)	60 (22.50)	χ2(4): 4.159	0.385
Diagnostic Delay (Yr)	0.73 (0.62)	0.34 (0.22)	1.51 (1.48)	0.43 (0.27)	χ2(3): 47.322	<0.001
**Md (95% CI)**						
Survival Time (M)	28.93 (24.00–33.86)	32.43 (24.75–40.11)	114.43 (71.02–157.84)	22.07 (7.72–36.42)	χ2(3): 11.130	0.011
**n (%)**						
Sex:						
Male	28 (58)	32 (67)	12 (55)	13 (62)	χ2: 1.184	0.757
King Stage:						
Stage 1	14 (29)	18 (38)	7 (32)	8 (38)	χ2(3): 1.025	0.795
Stage 2	17 (35)	12 (25)	7 (32)	8 (38)		
Stage 3	16 (33)	16 (33)	7 (32)	5 (24)		
Stage 4	1 (2)	2 (4)	1 (5)	0 (0)		
Onset type:						
Spinal	33 (69)	31 (65)	18 (82)	14 (67)	χ2: 2.185	0.535

**Table 2 cells-12-00933-t002:** Volume-of-interest (VOI)-based Mann–Whitney U test of ^18^F-FDG SUVR between *SOD1*-ALS (*n* = 22) and *C9orf72*-ALS patients; *n* = 48). FDR-corrected *p*-values < 0.05 for multiple testing were considered significant. Abbreviations: IQR, interquartile range; Md, median; sALS = sporadic ALS; SOD1-ALS = SOD1-associated ALS; SUVR = standardized uptake value ratio.

Median (IQR)	*SOD1*-ALS	*C9orf72*-ALS	Test Statistic	*p*_FDR_ Value
Mid Frontal gyrus	1.100 (0.07)	1.070 (0.07)	403.5	0.248
Precentral gyrus	1.015 (0.04)	0.990 (0.05)	272.0	0.007
Straight gyrus	0.975 (0.05)	0.970 (0.06)	479.5	0.711
Orbitofrontal cortex	1.000 (0.07)	1.010 (0.06)	485.0	0.746
Inferior frontal gyrus	1.075 (0.06)	1.050 (0.06)	373.0	0.181
Superior frontal gyrus	1.010 (0.05)	0.990 (0.04)	384.0	0.165
Medial orbital gyrus	0.960 (0.05)	0.970 (0.07)	442.5	0.490
Lateral orbital gyrus	1.025 (0.05)	1.020 (0.06)	479.0	0.732
Posterior orbital gyrus	0.940 (0.03)	0.960 (0.06)	452.0	0.515
Subgenual frontal cortex	0.830 (0.05)	0.835 (0.08)	509.0	0.855
Subcallosal area	0.825 (0.05)	0.800 (0.10)	427.0	0.389
Pre-subgenual frontal cortex	1.030 (0.06)	1.005 (0.12)	515.0	0.893
Hippocampus	0.770 (0.18)	0.780 (0.08)	486.5	0.739
Amygdala	0.720 (0.07)	0.745 (0.07)	379.5	0.168
Anterior temporal lobe (med)	0.740 (0.04)	0.755 (0.04)	398.5	0.229
Lateral Anterior temporal lobe	0.860 (0.06)	0.870 (0.04)	505.5	0.896
Parahippocampal and ambient gyri	0.780 (0.03)	0.790 (0.03)	416.0	0.317
Superior temporal gyrus, posterior	1.010 (0.08)	1.000 (0.06)	469.5	0.652
Middle and inferior temporal gyrus	0.940 (0.04)	0.940 (0.04)	506.0	0.875
Fusiform gyrus	0.840 (0.04)	0.840 (0.06)	506.0	0.875
Posterior temporal lobe	1.000 (0.03)	1.010 (0.03)	457.5	0.543
Superior temporal gyrus, anterior	0.840 (0.05)	0.820 (0.05)	449.5	0.537
Postcentral gyrus	1.010 (0.03)	0.980 (0.04)	246.5	0.006
Superior parietal gyrus	1.090 (0.05)	1.070 (0.04)	374.5	0.172
Inferolateral remainder parietal lobe	1.055 (0.04)	1.040 (0.04)	383.5	0.174
Lateral remainder of the occipital lobe	1.100 (0.05)	1.120 (0.07)	438.0	0.468
Lingual gyrus	1.180 (0.06)	1.190 (0.12)	450.5	0.524
Cuneus	1.220 (0.07)	1.255 (0.12)	487.0	0.720
Thalamus	0.935 (0.06)	0.880 (0.10)	278.5	0.012
Insula	0.930 (0.04)	0.925 (0.03)	515.5	0.874
Cingulate gyrus, anterior	1.005 (0.06)	0.975 (0.08)	376.0	0.167
Cingulate gyrus, posterior	1.160 (0.05)	1.120 (0.05)	304.0	0.019
Cerebellum	0.960 (0.09)	1.025 (0.07)	264.0	0.008
Medulla	0.715 (0.10)	0.790 (0.07)	250.0	0.005
Midbrain	0.740 (0.09)	0.810 (0.08)	288.0	0.012
Pons	0.625 (0.06)	0.680 (0.08)	303.0	0.019
Lentiform nucleus	1.080 (0.07)	1.155 (0.08)	214.0	0.003

**Table 3 cells-12-00933-t003:** Volume-of-interest (VOI)-based Mann–Whitney U test of 18F-FDG SUVR between *C9orf72*-ALS (*n* = 48) patients and matched sporadic ALS (*^C9orf72^*^-matched^sALS; *n* = 48) patients. FDR-corrected *p*-values < 0.05 for multiple testing were considered significant. Abbreviations: C9orf72-ALS, C9orf72-associated ALS; IQR, interquartile range; Md, median; sALS = sporadic ALS; SUVR = standardized uptake value ratio.

Median (IQR)	*C9orf72*-ALS	*^C9orf72^*^-matched^sALS	Test Statistic	*p*_FDR_-Value
Mid Frontal gyrus	1.070 (0.07)	1.085 (0.06)	1070.0	0.675
Precentral gyrus	0.990 (0.05)	1.010 (0.04)	797.0	**0.048**
Straight gyrus	0.970 (0.06)	0.950 (0.08)	1377.0	0.242
Orbitofrontal cortex	1.010 (0.06)	1.000 (0.07)	1419.0	0.142
Inferior frontal gyrus	1.050 (0.06)	1.055 (0.06)	1041.5	0.550
Superior frontal gyrus	0.990 (0.04)	1.000 (0.06)	997.5	0.412
Medial orbital gyrus	0.970 (0.07)	0.960 (0.04)	1429.5	0.138
Lateral orbital gyrus	1.020 (0.06)	1.000 (0.07)	1321.0	0.377
Posterior orbital gyrus	0.960 (0.06)	0.945 (0.06)	1300.5	0.407
Subgenual frontal cortex	0.835 (0.08)	0.820 (0.08)	1343.0	0.314
Subcallosal area	0.800 (0.10)	0.810 (0.10)	1133.0	0.940
Pre-subgenual frontal cortex	1.005 (0.12)	1.015 (0.12)	1152.0	1.000
Hippocampus	0.780 (0.08)	0.770 (0.09)	1231.5	0.668
Amygdala	0.745 (0.07)	0.720 (0.07)	1368.0	0.261
Ant. Temporal lobe (med)	0.755 (0.04)	0.730 (0.06)	1497.0	0.263
Ant. Temporal lobe (lateral)	0.870 (0.04)	0.860 (0.05)	1320.5	0.362
Parahippocampal and ambient gyri	0.790 (0.03)	0.790 (0.05)	1272.5	0.513
Superior temporal gyrus, posterior	1.000 (0.06)	1.015 (0.06)	1008.5	0.414
Middle and inferior temporal gyrus	0.940 (0.04)	0.930 (0.03)	1362.0	0.182
Fusiform gyrus	0.840 (0.06)	0.830 (0.05)	1261.0	0.538
Posterior temporal lobe	1.010 (0.03)	1.010 (0.04)	1206.0	0.751
Superior temporal gyrus, anterior	0.820 (0.05)	0.810 (0.08)	1170.0	0.920
Postcentral gyrus	0.980 (0.04)	1.010 (0.05)	670.0	**0.007**
Superior parietal gyrus	1.070 (0.04)	1.085 (0.04)	774.0	0.031
Inferolateral remainder parietal lobe	1.040 (0.04)	1.050 (0.03)	922.5	0.241
Lateral remainder occipital lobe	1.120 (0.07)	1.100 (0.06)	1340.0	0.309
Lingual gyrus	1.190 (0.12)	1.190 (0.11)	1217.0	0.710
Cuneus	1.255 (0.12)	1.230 (0.09)	1227.0	0.673
Thalamus	0.880 (0.10)	0.930 (0.09)	724.5	**0.019**
Insula	0.925 (0.03)	0.920 (0.06)	1306.0	0.395
Cingulate gyrus, ant.	0.975 (0.08)	1.005 (0.09)	957.0	0.312
Cingulate gyrus, post.	1.120 (0.05)	1.150 (0.05)	659.0	**0.011**
Cerebellum	1.025 (0.07)	0.985 (0.07)	1482.5	0.062
Medulla	0.790 (0.07)	0.740 (0.07)	1602.5	**0.012**
Midbrain	0.810 (0.08)	0.790 (0.07)	1453.0	0.100
Pons	0.680 (0.08)	0.650 (0.07)	1421.5	0.148
Lentiform nucleus	1.155 (0.08)	1.110 (0.09)	1569.5	**0.015**

**Table 4 cells-12-00933-t004:** Volume-of-interest (VOI)-based Mann–Whitney U test of ^18^F-FDG SUVR between *SOD1*-ALS (*n* = 22) patients and matched sporadic ALS (*^SOD1^*^-matched^sALS; *n* = 21) patients. FDR-corrected *p*-values < 0.05 for multiple testing were considered significant. Abbreviations: IQR, interquartile range; Md, median; sALS = sporadic ALS; SOD1-ALS = SOD1-associated ALS; SUVR = standardized uptake value ratio.

Median (IQR)	*SOD1*-ALS	*^SOD1-^*^matched^ sALS	Test Statistic	*p*_FDR_ Value
Mid Frontal gyrus	1.100 (0.07)	1.090 (0.03)	249.5	1.000
Precentral gyrus	1.015 (0.04)	1.000 (0.04)	299.0	1.000
Straight gyrus	0.975 (0.05)	0.990 (0.06)	207.0	1.000
Orbitofrontal cortex	1.000 (0.07)	1.000 (0.06)	214.0	0.965
Inferior frontal gyrus	1.075 (0.06)	1.050 (0.06)	265.0	1.000
Superior frontal gyrus	1.010 (0.05)	1.010 (0.05)	221.0	0.933
Medial orbital gyrus	0.960 (0.05)	0.960 (0.05)	208.0	1.000
Lateral orbital gyrus	1.025 (0.05)	1.010 (0.06)	260.5	1.000
Posterior orbital gyrus	0.940 (0.03)	0.960 (0.05)	195.5	1.000
Subgenual frontal cortex	0.830 (0.05)	0.830 (0.05)	243.5	0.937
Subcallosal area	0.825 (0.05)	0.820 (0.07)	229.0	0.988
Pre-subgenual frontal cortex	1.030 (0.06)	1.040 (0.07)	176.0	1.000
Hippocampus	0.770 (0.18)	0.770 (0.06)	221.0	0.905
Amygdala	0.720 (0.07)	0.730 (0.05)	207.5	1.000
Anterior temporal lobe (med)	0.740 (0.04)	0.750 (0.05)	200.0	1.000
Lateral Anterior temporal lobe	0.860 (0.06)	0.870 (0.05)	222.5	0.910
Parahippocampal and ambient gyri	0.780 (0.03)	0.780 (0.05)	227.0	0.975
Superior temporal gyrus, posterior	1.010 (0.08)	1.010 (0.04)	219.5	0.930
Middle and inferior temporal gyrus	0.940 (0.04)	0.950 (0.06)	212.0	1.000
Fusiform gyrus	0.840 (0.04)	0.830 (0.05)	252.5	1.000
Posterior temporal lobe	1.000 (0.03)	1.020 (0.03)	210.5	1.000
Superior temporal gyrus, anterior	0.840 (0.05)	0.830 (0.06)	193.5	1.000
Postcentral gyrus	1.010 (0.03)	1.010 (0.04)	276.5	1.000
Superior parietal gyrus	1.090 (0.05)	1.070 (0.06)	286.0	1.000
Inferolateral remainder parietal lobe	1.055 (0.04)	1.050 (0.03)	278.0	1.000
Lateral remainder of the occipital lobe	1.100 (0.05)	1.100 (0.07)	247.5	0.908
Lingual gyrus	1.180 (0.06)	1.190 (0.07)	230.0	0.981
Cuneus	1.220 (0.07)	1.210 (0.07)	301.0	1.000
Thalamus	0.935 (0.06)	0.940 (0.07)	213.0	0.978
Insula	0.930 (0.04)	0.930 (0.05)	214.5	0.877
Cingulate gyrus, anterior	1.005 (0.06)	1.000 (0.09)	248.0	0.930
Cingulate gyrus, posterior	1.160 (0.05)	1.140 (0.07)	262.0	1.000
Cerebellum	0.960 (0.09)	0.980 (0.06)	171.5	1.000
Medulla	0.715 (0.10)	0.740 (0.09)	194.0	1.000
Midbrain	0.740 (0.09)	0.760 (0.06)	179.0	1.000
Pons	0.625 (0.06)	0.650 (0.05)	200.5	1.000
Lentiform nucleus	1.080 (0.07)	1.100 (0.08)	203.4	1.000

## Data Availability

Not applicable.

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
