# Peer review of "Differences in Cerebral Glucose Metabolism in ALS Patients with and without *C9orf72* and *SOD1* Mutations"

_cells, 2023, doi:10.3390/cells12060933_

Round 1
Reviewer 1 Report
De Vocht et al. investigated the brain FDG-PET metabolic difference between different genetic groups of ALS, mainly C9ORF72 vs SOD1 vs sporadic ALS. No significant differences in cerebral glucose metabolism were found between SOD1-ALS and sporadic ALS matched, on the contrary patients with C9ORF72 showed significant differences in FDG uptake in the peri-rolandic region, thalamus and brainstem compared with patients with sporadic ALS matched. While this topic is of potential interest, I found several important concerns that significantly affect the impact of the study,
Major
Onset type: any metabolic difference between bulbar vs spinal?
Lack of clinical correlates: there is no evaluation of cognitive functioning as well as upper motor neuron burden through dedicated scales. Furthermore, there is no information on the clinical phenotypes. Without such clinical data, the study results remain largely speculative.
Matching Procedure: The SOD1-ALS patient group does not match the sporadic ALS matched group in terms of age, survival, and diagnostic delay. It would have been more interesting to compare SOD1 patients with an ALS population with slower disease progression (LMN variants?).
SOD1 group: I think the within-group heterogeneity is a big limitation of the study, it would also have been interesting to compare the SOD1 group vs the C9ORF72 group.
Minor
Table 1: diagnostic delay appears to be expressed in years and not in months.
Author Response
De Vocht et al. investigated the brain FDG-PET metabolic difference between different genetic groups of ALS, mainly C9ORF72 vs SOD1 vs sporadic ALS. No significant differences in cerebral glucose metabolism were found between SOD1-ALS and sporadic ALS matched, on the contrary patients with C9ORF72 showed significant differences in FDG uptake in the peri-rolandic region, thalamus and brainstem compared with patients with sporadic ALS matched. While this topic is of potential interest, I found several important concerns that significantly affect the impact of the study.
We thank the reviewer for their comments, as this valuable feedback improved the quality of the work.
MAJOR
Onset type: any metabolic difference between bulbar vs spinal?In line with the study of Sala et al (2019), we found no significant differences in cerebral glucose metabolism at puncorr < .001 at voxel level, and pFWE < .05 at cluster level in our cohort of sporadic bulbar and spinal ALS patients (including age and sex as nuisance variables).
Lack of clinical correlates:
there is no evaluation of cognitive functioning as well as upper motor neuron burden through dedicated scales. Furthermore, there is no information on the clinical phenotypes. Without such clinical data, the study results remain largely speculative.
We are indeed reporting on retrospective data and agree that the absence of cognitive measures for a large number of patients is a limitation to the study. Despite lacking a semi-quantitative measure of upper motor neuron burden through a dedicated scale, we did include upper motor neuron burden according to clinical and neurological evaluation.
Matching Procedure:
The SOD1-ALS patient group does not match the sporadic ALS matched group in terms of age, survival, and diagnostic delay. It would have been more interesting to compare SOD1 patients with an ALS population with slower disease progression (LMN variants?).
Our choice to favour age and disease stage over disease duration, when constructing artificial “control sporadic ALS groups” matched to our SOD1 and C9orf72 ALS cohort, was informed by recent studies by Canosa et al. (2021) and Ferraro et al. (2022), that demonstrated that hypo- and hypermetabolism on FDGPET correlates with disease stage and ageing, rather than disease progression rate.
We only found a significant difference in survival between C9orf72-ALS and SOD1-ALS (p5). As both genetic cohorts were unmatched and are associated with different disease progression rates, we anticipated this difference.
We indeed chose to include age at PET, king disease stage, sex, scanner type, diagnostic delay and extent of motor neuron involvement as matching variables in our propensity score matching model. Comparing the SOD1-ALS patient group to a cohort of slow progressors in our sporadic ALS cohort, did not enable us to match for all other relevant variables (i.e. age at PET, extent of motor neuron involvement, king disease stage). Despite the difference in diagnostic delay, our model proposed the best fit for each genetic carrier, considering all matching variables and associated multivariate interactions, unlike biserial matching.
SOD1 group:
I think the within-group heterogeneity is a big limitation of the study, it would also have been interesting to compare the SOD1 group vs the C9ORF72 group.
We agree that the within-group heterogeneity requires large multicentre approaches. We found a significantly higher FDG uptake in the C9orf72-ALS cohort in the brainstem and cerebellum and reduced FDG uptake in the basal ganglia in the SOD1-ALS cohort. These findings were added to the manuscript.
MINOR
Table 1: diagnostic delay appears to be expressed in years and not in months.
We corrected this error accordingly.
Reviewer 2 Report
The results are oddly undistinguished - little real difference is observed between the groups chosen for comparison. This I either because the groups are inappropriate or because the disease had different survival and progression rates different in the different classified groups; i.e, were at different stages and had different rates of progression, and that these factors were driving the different glucose metabolism results. However, since mitochondrial abnormalities have been invoked in C9 disease, mitochondrial dysmetabolism might be a factor - the authors shy away from this notion.
So the question arises as to whether these findings actually have significance in understanding the disease? It is worrisome that the authors suggest larger studies are be necessary - not a suggestion that inspires confidence. The crucial question as to the meaning of variabilities in progression rate and syndromology in different cases is not resolved by this work or even closely addressed. There do not seem to be any questions capable of being addressed as a result of the work described that have not already been mulled over by other investigators, largely without answers.
One can only conclude that the methodology employed was not capable of addressing the important underlying questions.
As an aside the description of patients as having European or Caucasian ancestry is surely outmoded? The notion of a Caucasian ancestry is far too simplistic and it's far better simply to refer to people as European - which of course allows for North African, Arab and even African cross relationships as well as Muslim, and central Asian genes.
The authors should be more honest in conceding that their data do not add substantively to what is known.
Author Response
We thank the reviewer for their comments, as this valuable feedback improved the quality of the work.
The results are oddly undistinguished - little real difference is observed between the groups chosen for comparison. This I either because the groups are inappropriate or because the disease had different survival and progression rates different in the different classified groups; i.e, were at different stages and had different rates of progression, and that these factors were driving the different glucose metabolism results.
We chose to include age at PET, king disease stage, sex, scanner type, diagnostic delay and extent of motor neuron involvement as matching variables in our propensity score matching model. Our choice to favour age and disease stage over disease duration, when constructing artificial “control sporadic ALS groups” matched to our SOD1-ALS and C9orf72-ALS cohort, was informed by recent studies by Canosa et al. (2021) and Ferraro et al. (2022), which demonstrated how these factors drive cerebral glucose metabolic changes. Our model thus proposed the best fit for each genetic carrier, considering all aforementioned matching variables and associated multivariate interactions.
However, since mitochondrial abnormalities have been invoked in C9 disease, mitochondrial dysmetabolism might be a factor - the authors shy away from this notion.
We had indeed not discussed the dysfunction of mitochondria as a potential factor to explain the glucose metabolic changes observed in C9orf72-ALS in our paper. A recent study by Terada et al. (2020) in Alzheimer’s Disease, revealed that mitochondria-related energy failure may precede hypometabolism and can be linked to a wide range of neurodegenerative processes.
We added the following segment to the discussion: “The clusters of significantly reduced glucose metabolism in the peri-rolandic region, extending to the precuneus, and thalamus suggest that both the thalamus and peri-rolandic region play a key role in C9orf72-mediated disease [26-28, 37]. Reduced FDG uptake in C9orf72-ALS can reflect multiple neurobiological underpinnings, such as reduced excitability [38], synapse loss [39], neurodegeneration. Mitochondrial dysfunction [40] may also contribute to the observed changes, as a recent study employing the [18F]2-tert- butyl-4-chloro-5-2H- pyridazin-3-one (BCPP-EF) radioligand for mitochondrial complex demonstrated that mitochondrial complex I reduction can be linked to a range of downstream neurodegenerative processes, such as hypometabolism. Follow-up PET studies with [18F]BCPP-EF and [18F]FDG, would enable in vivo investigation of dysfunction of mitochondria and its relation to hypometabolism in C9orf72-ALS [41].”
So the question arises as to whether these findings actually have significance in understanding the disease? It is worrisome that the authors suggest larger studies are be necessary - not a suggestion that inspires confidence.
Given the large number of different mutations observed in SOD1-ALS, this results in large within-group heterogeneity. For that purpose, we are convinced that large multicentre approaches are needed to gain a better understanding of the glucose metabolic changes associated with different SOD1 variants.
The crucial question as to the meaning of variabilities in progression rate and syndromology in different cases is not resolved by this work or even closely addressed. There do not seem to be any questions capable of being addressed as a result of the work described that have not already been mulled over by other investigators, largely without answers. One can only conclude that the methodology employed was not capable of addressing the important underlying questions.
The aim of our study was to identify glucose metabolic changes associated with genetic variants of ALS, in comparison with matched sporadic ALS patients. We believe that we were able to address this objective research question with the methodology used in the study.
As an aside the description of patients as having European or Caucasian ancestry is surely outmoded? The notion of a Caucasian ancestry is far too simplistic and it's far better simply to refer to people as European - which of course allows for North African, Arab and even African cross relationships as well as Muslim, and central Asian genes.
We agree with the reviewer and have adjusted the manuscript text accordingly.
The authors should be more honest in conceding that their data do not add substantively to what is known.
Given the conflicting findings from previous studies on glucose metabolic changes in C9orf72-ALS, as well as the limited number of PET studies in SOD1-ALS, we think that this research contributes to the field by revealing the crucial role of the brainstem in C9orf72-ALS, and the presence of widespread glucose metabolic changes in SOD1-ALS.
Reviewer 3 Report
If it is not possible to implement the results with data from other genetic populations, the title of the paper should be changed as it is misleading: the authors only succeed in analysing cohorts of patients who presented the SOD1 mutation or the C9orf72 hexanucleotide expansion.
Not only that, for the SOD1 gene alone there are more than 180 different mutations, it would be interesting to be able to identify the prevalence of these mutations and observe changes in glucose metabolism.
It would be interesting to observe the results obtained in the form of paired graphs showing the correlation between the uptake in the regions of each group and/or density graphs showing the frequency of patients with low or high uptake of 18F-FDG for each group.
Line 29: write FDGPET in full as this is the first time it appears
Ref 51: typo in the title
Author Response
We thank the reviewer for their comments, as this valuable feedback improved the quality of the work.
If it is not possible to implement the results with data from other genetic populations, the title of the paper should be changed as it is misleading: the authors only succeed in analysing cohorts of patients who presented the SOD1 mutation or the C9orf72 hexanucleotide expansion.
We agree to this change and adjusted the title accordingly to: “Differences in cerebral glucose metabolism in ALS patients with and without C9orf72 and SOD1 mutations.”
Not only that, for the SOD1 gene alone there are more than 180 different mutations, it would be interesting to be able to identify the prevalence of these mutations and observe changes in glucose metabolism.
We thank the reviewer for this comment and are convinced that large multicentre approaches are indeed needed to gain a better understanding of the glucose metabolic changes associated with different SOD1 variants.
It would be interesting to observe the results obtained in the form of paired graphs showing the correlation between the uptake in the regions of each group and/or density graphs showing the frequency of patients with low or high uptake of 18F-FDG for each group.
We agree with the reviewer and added a supplementary figure (Figure S3) on p28, detailing the FDG uptake in different ALS groups in key Volumes-of-interest.
Line 29: write FDGPET in full as this is the first time it appears
We corrected this error accordingly.
Ref 51: typo in the title
We corrected this error accordingly.
Round 2
Reviewer 1 Report
I have no further comments.
Author Response
We would like to thank the reviewer for their previous comments and suggestions.
Reviewer 2 Report
An improved version, that merits publication.
As a native English speaker I find the notion of 'increased hypometabolism' strange, even laughable (Introduction). Could the authors be persuaded to recognise that one cannot increase a negative quantity? (In any language). Hypometabolism could be further decreased etc. My English master at school would have refused to read any further!
Author Response
We changed the wording of the manuscript for clarity.
As a native English speaker I find the notion of 'increased hypometabolism' strange, even laughable (Introduction). Could the authors be persuaded to recognise that one cannot increase a negative quantity? (In any language). Hypometabolism could be further decreased etc. My English master at school would have refused to read any further!
We changed the wording on lines 37, 38, 270, 325, 326 and 385:
P1, line 37: excess hypometabolism -> more pronounced hypometabolism
P1, line 38: excess hypermetabolism -> hypermetabolism
P7, line 270: Excess hypermetabolism -> Hypermetabolism
P15, line 325: excess hypometabolism -> more pronounced hypometabolism
P15, line 326: excess hypermetabolism -> hypermetabolism
P16, line 385: excess hypermetabolism -> significant hypermetabolism
Reviewer 3 Report
The authors replied to all comments
Author Response

(The authors gave the same response as above.)
